# Reaction-induced rheological weakening enables oceanic plate subduction

Ken-ichi Hirauchi[1], Kumi Fukushima[2], Masanori Kido[3], Jun Muto[3] & Atsushi Okamoto[4]

Earth is the only terrestrial planet in our solar system where an oceanic plate subducts beneath an overriding plate. Although the initiation of plate subduction requires extremely weak boundaries between strong plates, the way in which oceanic mantle rheologically weakens remains unknown. Here we show that shear-enhanced hydration reactions contribute to the generation and maintenance of weak mantle shear zones at mid-lithospheric depths. High-pressure friction experiments on peridotite gouge reveal that in the presence of hydrothermal water, increasing strain and reactions lead to an order-of-magnitude reduction in strength. The rate of deformation is controlled by pressure-solution-accommodated frictional sliding on weak hydrous phyllosilicate (talc), providing a mechanism for the 'cutoff' of the high peak strength at the brittle-plastic transition. Our findings suggest that infiltration of seawater into transform faults with long lengths and low slip rates is an important controlling factor on the initiation of plate tectonics on terrestrial planets.

[1] Department of Geosciences, Faculty of Science, Shizuoka University, 836 Ohya, Suruga-ku, Shizuoka 422-8529, Japan. [2] Department of Science, Graduate School of Integrated Science and Technology, Shizuoka University, 836 Ohya, Suruga-ku, Shizuoka 422-8529, Japan. [3] Department of Earth Science, Graduate School of Science, Tohoku University, Sendai 980-8578, Japan. [4] Department of Environmental Studies for Advanced Society, Graduate School of Environmental Studies, Tohoku University, Sendai 980-8579, Japan. Correspondence and requests for materials should be addressed to K.-i.H. (email: hirauchi.kenichi@shizuoka.ac.jp).

Plate tectonics on Earth represents a unique style of mantle convection among terrestrial planets in our solar system[1,2]. In plate tectonics, an oceanic plate can subduct into the deep mantle beneath an overriding plate, thereby influencing the geodynamic and geochemical evolution of the Earth[2]. However, the underlying physics of subduction initiation remain poorly understood. Numerical models have demonstrated that a preexisting weak zone between strong plates is needed for subduction to be initiated[3–6]; the average strength of a weak fault that needs to be overcome by ridge push alone must be < 30 MPa (refs 7,8).

Figure 1 shows a strength profile for an incipient subduction thrust fault within 25-Myr-old oceanic lithosphere. If the brittle regime is dominated by frictional sliding on typical crustal and mantle rocks (that is, gabbro and peridotite) and the ductile regime is controlled by low-temperature plasticity (Peierls mechanism) of olivine at a strain rate of $10^{-15} s^{-1}$, then the differential stress at the brittle-ductile transition (BDT) is 1,220 MPa at a depth of 14 km, which is at least one order-of-magnitude higher than the average fault strength mentioned above. Field and experimental studies have shown that for polymineralic (olivine and pyroxene) mantle rocks, the transition from dislocation creep to diffusion creep results in rheological weakening and subsequent strain localization (Fig. 1)[9–11]. However, the activation of diffusion creep is also strongly dependent on temperature and does not contribute to a significant reduction in the high stresses near the BDT (Fig. 1).

This result indicates that a frictionally weak fault with a coefficient of friction ($\mu$) of < 0.1 must exist at depths shallower than 20 km (Fig. 1).

In intra-oceanic environments, changes in plate motion may convert existing transform faults and fracture zones into a self-sustaining subduction zone[4,5,12]. Although the oceanic lithosphere is considered to be dry, due to dehydration melting at mid-ocean ridges[2], infiltration of seawater into a preexisting fault zone will lead to the formation of hydrous minerals (for example, amphibole, serpentine and talc)[13,14]. In particular, serpentine and talc as phyllosilicates have much lower friction coefficients than anhydrous mantle minerals[15,16]. Therefore, frictional sliding on weak hydrous phyllosilicates may significantly contribute to the truncation of the maximum fault strength of oceanic lithosphere.

To test this hypothesis, we perform friction experiments designed to promote shear-enhanced hydration reactions of mantle peridotites at simulated $P–T$ conditions corresponding to mid-lithospheric depths. Our experimental and microstructural observations show that newly formed talc, one of the weakest hydrous phyllosilicates, induces strain localization and significantly reduces sample strength. Our findings suggest that deep-level water-rock interactions in a mantle shear zone may strongly influence the initiation of oceanic plate subduction on terrestrial planets.

## Results

**Mechanical data.** Simple-shear deformation experiments were conducted on hot-pressed aggregates of olivine (70%) and orthopyroxene (30%) of initial compositions $Fo_{91}$ and $En_{90}$, respectively (Table 1), at a temperature ($T$) of 500 °C and a confining pressure ($P_c$) of 1.0 GPa, under hydrothermal

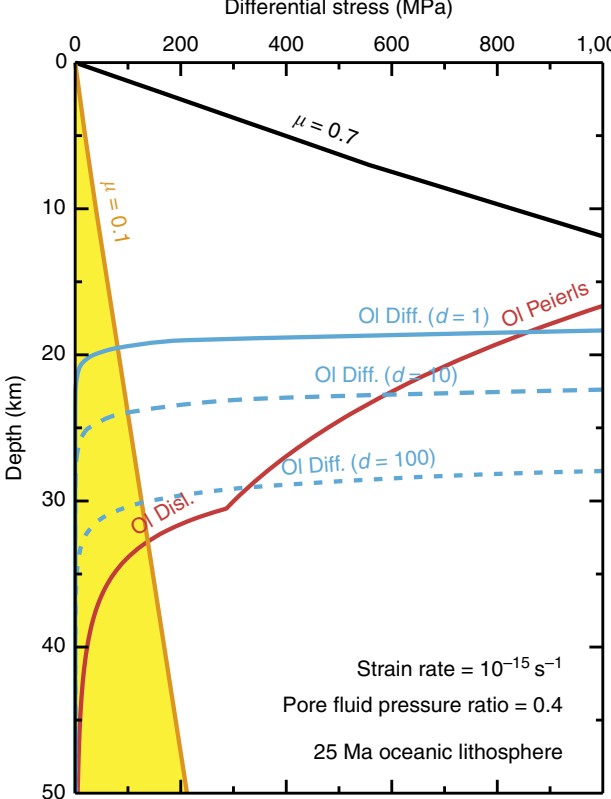

**Figure 1 | Strength profile for a subduction thrust fault developing in 25-Myr-old oceanic lithosphere.** Stress in the brittle regime is predicted by frictional sliding with a friction coefficient ($\mu$) of 0.1 (orange) and 0.7 (black), and hydrostatic pore fluid pressure. Data of dry olivine (ol) are taken from flow laws for diffusion creep (blue) and dislocation creep (power law; red) and Peierls mechanism (red) at a strain rate of $10^{-15} s^{-1}$, given by Hirth and Kohlstedt[42] and Mei et al.[29]. For diffusion creep, we assumed a shear zone with grain sizes ($d$) of 1, 10, or 100 μm.

### Table 1 | Chemical composition of minerals.

| Run no. | Starting material | | KF-08 | | |
|---|---|---|---|---|---|
| **Mineral** | ol | opx | ol_core | ol_rim | tlc |
| **Pts.** | 10 | 8 | 11 | 7 | 7 |
| $SiO_2$ | 39.8 | 56.5 | 41.3 | 41.1 | 59.0 |
| $TiO_2$ | 0.02 | 0.02 | 0.02 | 0.03 | 0.02 |
| $Al_2O_3$ | 0.03 | 0.17 | 0.02 | 0.03 | 0.38 |
| $FeO^*$ | 8.85 | 6.85 | 8.62 | 11.8 | 2.42 |
| $MnO$ | 0.14 | 0.41 | 0.12 | 0.33 | 0.02 |
| $Cr_2O_3$ | 0.03 | 0.01 | 0.03 | 0.01 | 0.01 |
| $MgO$ | 49.2 | 34.8 | 51.0 | 48.0 | 29.8 |
| $CaO$ | 0.08 | 0.17 | 0.08 | 0.06 | 0.10 |
| $Na_2O$ | 0.02 | 0.04 | 0.02 | 0.09 | 0.51 |
| $K_2O$ | 0.01 | 0.00 | 0.01 | 0.01 | 0.04 |
| $NiO$ | – | 0.01 | – | – | – |
| Total | 98.2 | 99.0 | 101.2 | 101.4 | 92.4 |
| *Cations* | *O = 4* | *O = 6* | *O = 4* | *O = 4* | *O = 11* |
| Si | 0.99 | 1.98 | 0.99 | 1.00 | 3.92 |
| Ti | 0.00 | 0.00 | 0.00 | 0.00 | 0.00 |
| Al | 0.00 | 0.00 | 0.00 | 0.00 | 0.01 |
| Fe | 0.18 | 0.20 | 0.17 | 0.24 | 0.13 |
| Mn | 0.00 | 0.01 | 0.00 | 0.01 | 0.00 |
| Cr | 0.00 | 0.00 | 0.00 | 0.00 | 0.00 |
| Mg | 1.82 | 1.82 | 1.83 | 1.74 | 2.95 |
| Ca | 0.00 | 0.01 | 0.00 | 0.00 | 0.01 |
| Na | 0.00 | 0.00 | 0.00 | 0.00 | 0.07 |
| K | 0.00 | 0.00 | 0.00 | 0.00 | 0.00 |
| Ni | – | 0.00 | – | – | – |
| Sum | 3.01 | 4.02 | 3.00 | 3.00 | 7.09 |
| Mg#† | 90.8 | 90.1 | 91.3 | 87.9 | 95.6 |

ol, olivine; opx, orthopyroxene; Pts, number of points analyzed; tlc, talc.
*Total iron given as FeO.
†Mg# = Mg/(Mg + Fe).

**Table 2 | Experimental conditions and the results.**

| Run no. | Sample | Pressure (GPa) | Temperature (°C) | Water added (wt%) | Heating time (h)* | Deformation time (h) | Total shear strain ($\gamma$) | Shear strain rate ($s^{-1}$) | Initial peak shear stress (MPa)† | Shear stress at $\gamma = 4.0$ (MPa) |
|---|---|---|---|---|---|---|---|---|---|---|
| KF-02 | 70% ol, 30% opx | 1.0 | 500 | 4.0 | 26 | 94 | 4.7 | $6.5 \times 10^{-6}$ | 202 (0.6) | 157 |
| KF-03 | 70% ol, 30% opx | 1.0 | 500 | 4.0 | 16 | 81 | 5.2 | $8.1 \times 10^{-6}$ | 379 (0.8) | 83 |
| KF-06 | 70% ol, 30% opx | 1.0 | 500 | 4.0 | 1 | 3 | 3.7 | $2.2 \times 10^{-4}$ | 171 (0.9) | 208‡ |
| KF-07 | 70% ol, 30% opx | 1.0 | 500 | 4.0 | 17 | 31 | 4.5 | $1.9 \times 10^{-5}$ | 326 (0.7) | 124 |
| KF-08 | 70% ol, 30% opx | 1.0 | 500 | 4.0 | 12 | 12 | 4.5 | $5.4 \times 10^{-5}$ | 324 (0.7) | 245 |
| KF-09 | 70% ol, 30% opx | 1.0 | 500 | 4.0 | 11 | 12 | 1.1 | $1.8 \times 10^{-5}$ | 359 (0.9) | NA |
| KF-10 | 70% ol, 30% opx | 1.0 | 500 | 4.0 | 21 | 147 | 4.4 | $4.3 \times 10^{-6}$ | 383 (0.7) | 39 |
| KF-11 | 70% ol, 30% opx | 1.0 | 500 | 4.0 | 13 | 24 | 2.7 | $1.8 \times 10^{-5}$ | 411 (0.8) | NA |
| KF-12 | 70% ol, 30% opx | 1.0 | 500 | 4.0 | 20 | NA | NA | NA | NA | NA |
| KF-13 | 70% ol, 30% opx | 1.0 | 500 | 4.0 | 168 | NA | NA | NA | NA | NA |
| KF-14 | 100% opx | 1.0 | 500 | 4.0 | 36 | 132 | 3.6 | $4.3 \times 10^{-6}$ | 322 (0.9) | 218‡ |

*Heating time includes the duration of a hydrostatic experiment or the time required for the piston to hit the sample in a deformation experiment.
†Shear strain of initial peak stress is given in parentheses.
‡Shear stress value was extrapolated to $\gamma$ of 4.0.

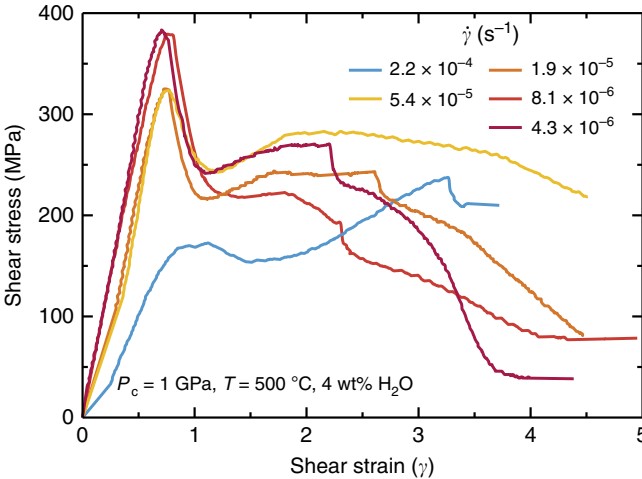

**Figure 2 | Shear stress–shear strain curves for olivine-orthopyroxene samples.** The samples were sheared at a confining pressure of 1 GPa, a temperature of 500 °C, and shear strain rates ($\gamma$) between $10^{-4}$ and $10^{-6}\,s^{-1}$, under hydrothermal conditions.

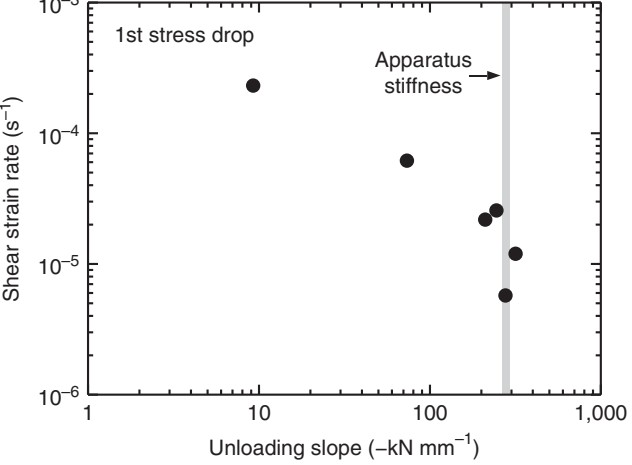

**Figure 3 | Characteristics of the first stress drop event in each deformation experiment.** Unloading slope ($-\Delta f/\Delta x$) values are plotted as a function of shear strain rate. The grey bar represents the approximate apparatus stiffness ($-\Delta f/\Delta x = \sim 280\,\text{kN mm}^{-1}$).

conditions (Supplementary Figs 1 and 2). The powdered sample, with 4 wt% deionized water added, was sandwiched between two alumina pistons, cut at 45° with respect to the direction of maximum compression, producing a ~0.6-mm-thick layer of simulated fault gouge after the application of confining pressure (before shearing). After annealing at $T = 500\,°C$ for 11–21 h, the samples were deformed to shear strains ($\gamma$) of 1.1–5.2 at a constant shear strain rate ($\dot{\gamma}$) ranging from $2.2 \times 10^{-4}$ to $4.3 \times 10^{-6}\,s^{-1}$. Details of the experimental conditions and key mechanical data are described in Table 2.

Figure 2 shows shear stress ($\tau$) versus shear strain curves for the olivine-orthopyroxene aggregates deformed at different shear strain rates ($\dot{\gamma} = 10^{-4}$ to $10^{-6}\,s^{-1}$). At the highest strain rate ($\dot{\gamma} = 2.2 \times 10^{-4}\,s^{-1}$), the sample showed strain hardening after a low initial peak stress (170 MPa). In contrast, at lower strain rates ($\dot{\gamma} < 10^{-4}\,s^{-1}$) the samples exhibited a high initial peak stress followed by a large stress drop, and then weakened with increasing strain. The sudden stress drops resulted from dynamic stick-slip, as the slope of the force–displacement curves ($-\Delta f/\Delta x$) during stress drop is close to that of the apparatus stiffness ($-\Delta f/\Delta x = \sim 280\,\text{kN mm}^{-1}$; Fig. 3). At $\dot{\gamma} < 10^{-4}\,s^{-1}$, the initial peak stresses are largely independent of strain rate ($\tau = 300$–400 MPa), while those at $\gamma = 4$ tend to gradually decrease from 250 to 40 MPa with decreasing strain rate (Fig. 4).

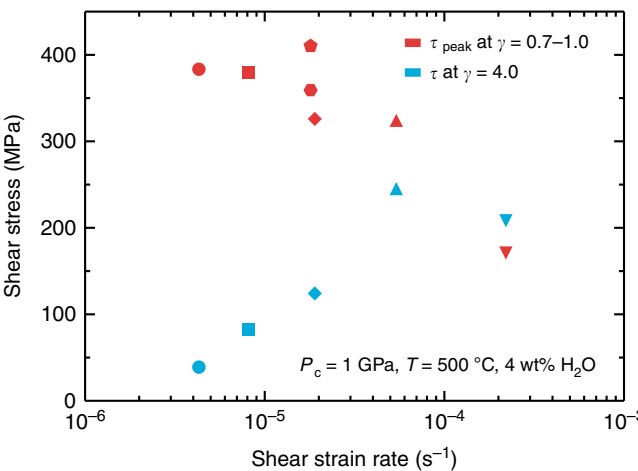

**Figure 4 | Dependence of shear stress in olivine-orthopyroxene samples on shear strain rate.** The shear stress values are those observed at the initial peak (red) and at a shear strain of four (blue). Each symbol represents the value for an individual experiment.

**Microstructural observations.** Figure 5 shows scanning electron microscope (SEM) images of the olivine-orthopyroxene aggregates deformed to shear strains between 1.1 and 4.5. At the onset of unstable slip ($\gamma = 1.0$; Supplementary Fig. 3), the sample contains narrow ($<15\,\mu m$) localized shear zones of boundary (B) and Riedel ($R_1$) shears[17], which occur directly adjacent to the piston–sample boundary and are oriented at $\sim 25°$ synthetic to the sinistral shear zone, respectively (Fig. 5a). The incipient shear zones are characterized by grain size reduction caused by cracking and comminution (Fig. 5c). With increasing shear strain ($\gamma = 2.7 - 5.2$), shear deformation is accommodated by boundary-parallel Y shears in addition to B and $R_1$ shears (Fig. 5b). Intensified grain comminution leads to widening of the localized shears associated with talc, a newly formed hydrous phase (Figs 5d and 6). The basal plane of the talc grains is strongly aligned with the localized shear plane (Fig. 5e). Talc also appears to grow at the expense of orthopyroxene (Fig. 5f), which exhibits irregular and serrated grain boundaries indicative of pressure solution[18].

**Discussion**
Microstructural and chemical analyses suggest that the hydration reaction proceeds via a process under the experimental conditions, as follows:

$$5(Mg, Fe)SiO_3 + H_2O = (Mg, Fe)_3Si_4O_{10}(OH)_2 + (Mg, Fe)_2SiO_4$$

orthopyroxene + water = talc + olivine

$$(1)$$

The preferential dissolution of orthopyroxene occurs because orthopyroxene reacts more rapidly than olivine at temperatures above $300\,°C$ (ref. 19). Petrographic observations of oceanic peridotites along a fracture zone near the Mid-Atlantic Ridge reveal evidence of the preferential replacement of orthopyroxene by talc at temperatures above $350\,°C$ (ref. 13). Iron-rich rims in olivine grains (Supplementary Fig. 4) indicate that excess iron, released during the formation of talc, is accommodated by newly crystallized Fe-rich olivine that overgrows the pre-existing olivine (Table 1).

The talc preferentially forms within localized shear zones ($R_1$, B and Y) where cataclasis-related grain size reduction is most intense. The grain size reduction is expected to enhance the preferential growth of talc within such fine-grained zones, as both

diffusive mass transfer and chemical reactions are enhanced for smaller grain sizes[20]. Furthermore, since strain is likely to be localized within the weak talc-bearing zones, talc will continue to grow preferentially in the developing shear zones through a positive feedback mechanism, resulting in widening of the talc-rich zone with increasing shear strain.

At $\dot\gamma < 10^{-4}\,s^{-1}$, the average proportion of talc formed along the localized shears increases only slightly (from 25 to 30%; Fig. 7) with decreasing shear strain rate, despite reaction times that differ by a factor of seven (24 versus 168 h). We also find that larger shear strains lead to a greater amount of reaction products, despite a total heating time that is almost constant ($\sim 24\,h$; compare Fig. 5a with Fig. 5b). These results indicate that the reaction rate of the fault zone material depends on the amount of strain that accumulates during cataclastic deformation, rather than time. In addition, at $\dot\gamma < 10^{-4}\,s^{-1}$ the steady-state shear strength shows a positive dependence on shear strain rate, suggesting the operation of frictional-viscous flow[21] controlled by frictional sliding on weak phyllosilicates (for example, talc) accommodated by pressure solution of the rigid clasts (for example, olivine and orthopyroxene); otherwise, rate-insensitive cataclastic flow becomes dominant at higher strain rate (that is, $\dot\gamma = 2.2 \times 10^{-4}\,s^{-1}$; Fig. 4). At lower strain rates, accommodation of slip along (001) basal planes in talc occurs more readily due to pressure solution[21].

Talc is one of the weakest phyllosilicates, showing no dependence of $\mu$ on effective normal stress even at high pressures and temperatures[16]. The apparent values of the steady-state friction coefficient ($\mu = \tau/\sigma_n$, where $\sigma_n$ is normal stress) for the samples deformed at $\dot\gamma = 10^{-6}\,s^{-1}$ are 0.07–0.13, which is consistent with those for pure talc at an effective normal stress of $0.1\,GPa$ (0.10–0.15 at $T = 200–400\,°C$)[16]. This indicates that the development of an interconnected network of phyllosilicates weakens the composite layer to a frictional strength similar to that of the phyllosilicate itself, which is consistent with previous findings[20,22]. Furthermore, our experiments suggest that the addition of even a small amount of water that is not enough to cause complete hydration, and subsequent strain accumulation ($\gamma > \sim 1$) leads to rheological weakening of the mantle shear zone within the oceanic lithosphere.

The minimum value of steady-state shear stress observed in our experiments (that is, at $\dot\gamma = 4.3 \times 10^{-6}\,s^{-1}$) was around $40\,MPa$ (Fig. 2), which is slightly higher than the average shear stress required for subduction initiation on an incipient thrust fault[7,8]. However, friction experiments have shown that the steady-state friction coefficient ($\mu_{ss}$) of talc has a positive dependence on sliding velocity $V$ ($\Delta\mu_{ss}/\Delta\ln V = 0.001 - 0.008$ at $400\,°C$)[16], suggesting a further decrease in stress at natural strain rates. It is also important to note that thermally activated processes, such as dislocation and diffusion creep, become active at depths below the BDT (Fig. 1), thereby strongly reducing the plastic strength with increasing temperature.

The species of hydrous minerals formed by the hydration of mantle peridotite is dependent on the amount of cations dissolved in water, and also on the pressure and temperature conditions[23]. The minimum value of differential stress observed in our experiments (Fig. 2) may provide a lower bound for the strength of partially hydrated mantle rocks at the same pressure, as talc is frictionally weaker than other hydrous minerals (for example, amphibole and serpentine)[15,24,25]. However, previous high-pressure ($>1\,GPa$) deformation experiments have shown that lizardite, a low-temperature serpentine phase that is stable below $300\,°C$ (refs 26,27), deforms by plastic flow controlled by dislocation glide on its basal plane, which gives a flow stress of $\sim 100\,MPa$ (ref. 28). This indicates that the onset of plastic deformation in serpentine has the potential to weaken shallow

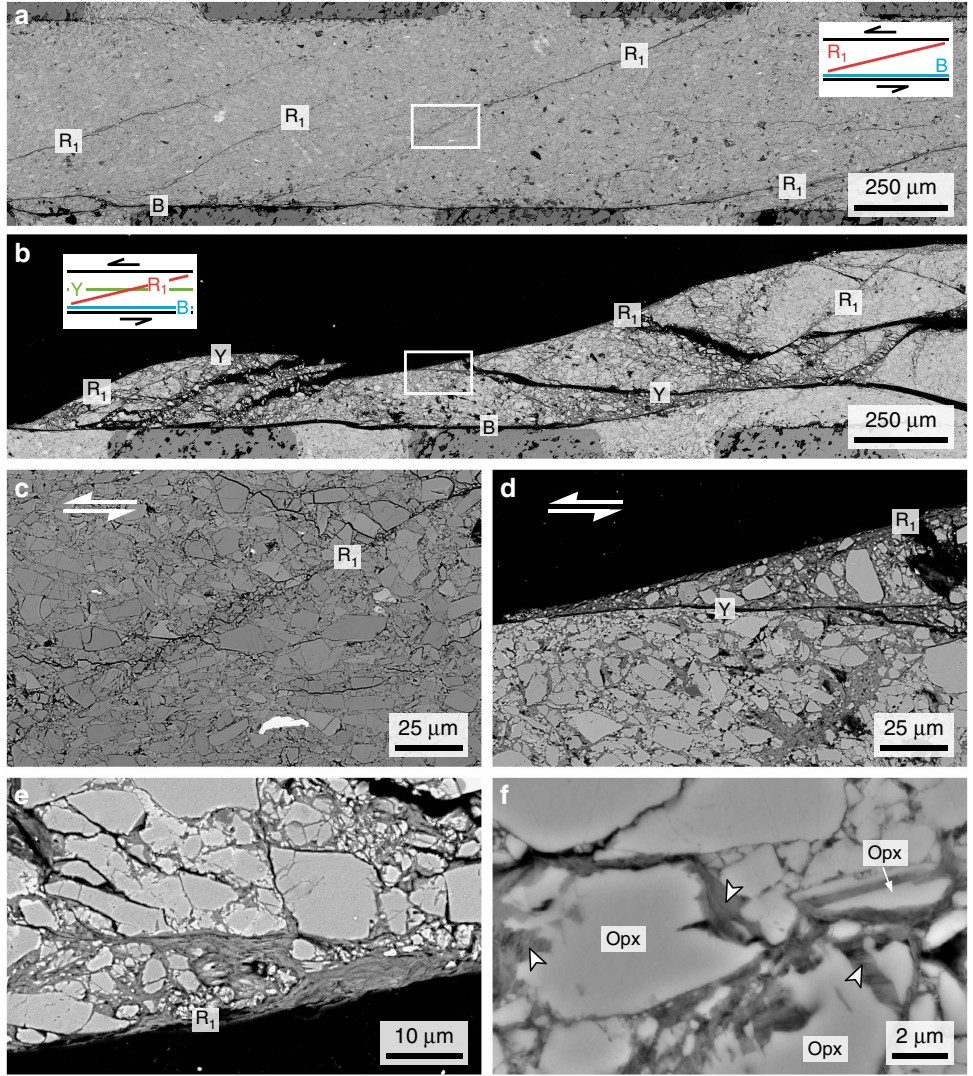

**Figure 5 | Back-scattered electron images of deformed olivine-orthopyroxene samples. (a)** Specimen deformed to a shear strain ($\gamma$) of 1.1 (Run KF-09). Note the incipient development of localized shear zones of boundary (B) and Riedel ($R_1$) shears within the gouge layer. **(b)** Run KF-08 ($\gamma = 4.5$). Note the well-developed localized shears (B, $R_1$ and Y) in which grain size reduction occurs within a wider zone. **(c)** Enlargement of the area in (**a**) indicated by the white rectangle. The thin incipient $R_1$ shear is characterized by grain size reduction. **(d)** Enlargement of the area in (**b**) indicated by the white rectangle. Intense grain size reduction along the localized shears results in the formation of talc (dark grey). **(e)** Run KF-03 ($\gamma = 5.2$). Note the alignment of platy talc grains (dark grey) along the $R_1$ shear plane. **(f)** Run KF-10 (heating time: 168 h). Note the preferential dissolution of orthopyroxene (opx), showing irregular and serrated grain boundaries. Talc is indicated by small arrows.

lithospheric shear zones[28]. At deeper levels ($>700$–$800\,^{\circ}$C), where all the hydrous phases become unstable, olivine and pyroxene deform by Peierls mechanism[29]; furthermore, grain size reduction by dynamic recrystallization in polymineralic domains may lead to a switch in deformation mechanism to diffusion creep, resulting in a marked reduction in strength (Fig. 1)[9–11].

Our experiments demonstrate that grain size reduction by cataclasis, hydration reactions, and subsequent pressure-solution-accommodated frictional sliding on phyllosilicates (that is, frictional-viscous flow) contribute to the initiation and maintenance of weak mantle shear zones within oceanic lithosphere. On the other hand, high pore fluid pressures[5,30] and shear heating[31] have been suggested as possible mechanisms that truncate the brittle-plastic strength envelope. In particular, an increase in pore fluid pressure ($P_f$) within a fault zone acts to reduce the effective normal stress ($\sigma_n^{\text{eff}} = \sigma_n - P_f$) and thereby lower the shear stress. Our observations may imply that the development of interconnected networks of phyllosilicates

(Fig. 5b,c), which have low permeabilities[32], allows high $P_f$ values to persist in the deep portions of oceanic faults (if the input of fluids is continuous).

The mechanism that introduces water into the upper mantle within oceanic lithosphere remains unclear, because preexisting weak zones (such as transform faults and fracture zones) are commonly limited to several kilometres beneath the seafloor[2,33,34]. However, this is not always the case: for example, in the present day, the long (900 km), low-slip-rate (2 cm yr$^{-1}$) Romanche Fault, a transform fault on the Mid-Atlantic Ridge, is characterized by seismicity that extends to depths of 20 km, indicating that the fault zone penetrates into the upper mantle[35]. For a half-space cooling model, the temperature at the centre of the fault at 20 km depth is $\sim500\,^{\circ}$C (ref. 35), which lies within the stability field of talc ($<700\,^{\circ}$C (ref. 36)). Geodynamic models have also predicted that the depths of brittle deformation and fluid circulation increase with decreasing geothermal gradient[37]. Therefore, we suggest that as

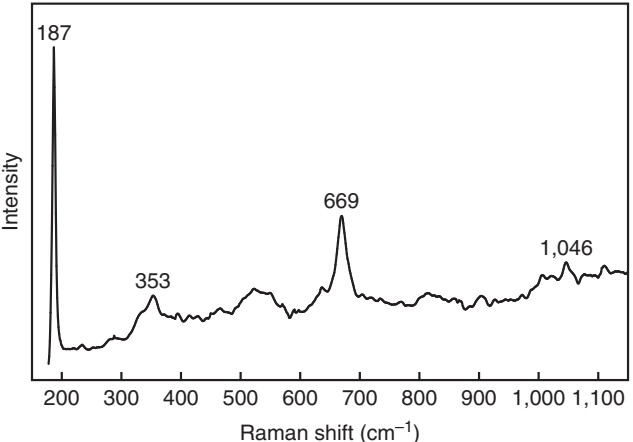

**Figure 6 | Spectra of reaction products.** Raman spectra of the reaction product formed in orthopyroxene aggregates (Run KF-14) show the peaks for talc.

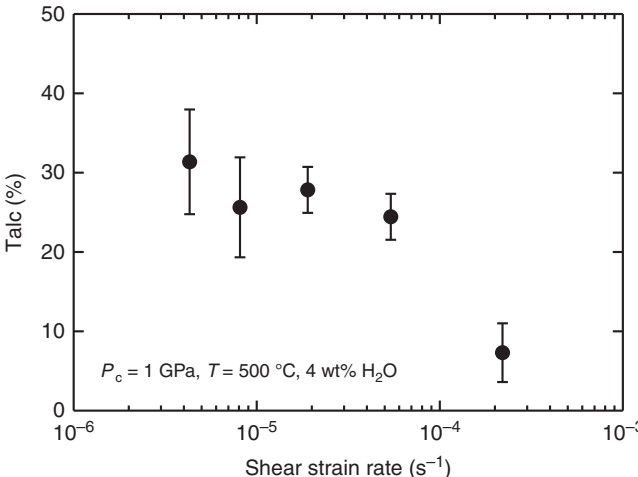

**Figure 7 | Reaction progress in olivine-orthopyroxene samples.** Average area proportion of talc (%) formed along localized shears (R$_1$, B and Y), plotted as a function of shear strain rate for olivine-orthopyroxene samples. Error bars indicate one standard deviation. The area fraction of talc within ~50 μm of the localized shears was deduced from image analysis, using back-scattered electron (BSE) imaging. From each BSE image (for example, Fig. 5d), we obtained the area fractions of pores (black), olivine and orthopyroxene (right grey), and talc (dark grey).

oceanic lithosphere ages, the depth extent of transform faults gradually increases via repeated brittle failures, resulting in the formation of phyllosilicate-rich peridotite fault gouge due to seawater infiltration and hydrothermal alteration.

Our study demonstrates that preexisting transform faults and fracture zones act as weak zones required for oceanic plate subduction to be initiated, as suggested by numerical models[4,5,12]. The conversion of such strike-slip fault zones to proto-subduction zones, caused by changes in plate motion, might have occurred at the Izu-Bonin-Mariana and Puysegur-Fiordland subduction zones[4,12,30,38]. The remaining question is, however, how transform-like plate boundaries could develop prior to the operation of plate tectonics. Indeed, regional strike-slip faults (as well as rifts), resulting from plume-related horizontal mantle flow, have been found on Venus[39]. Although the exact mechanism is still unclear, recent numerical models[6],

which incorporated no preexisting weak plate boundaries, showed that lithospheric damage promotes strain localization and weakening, eventually leading to the formation of strike-slip boundaries at Earth-like conditions. In the models[6], lithospheric damage is caused by the operation of grain-size sensitive diffusion creep in peridotite mylonites. Our results also imply that damage in the presence of hydrothermal seawater leads to the development of long-lived, weak hydrated mantle shear zones in cold lithosphere.

## Methods

**Starting materials.** The starting materials for olivine and orthopyroxene are single crystals from San Carlos, Arizona and from Tanzania, respectively. Representative chemical compositions of the olivine and orthopyroxene used in this study are listed in Table 1. The minerals were ground in a steel mortar and sieved to obtain a grain size less than 20 μm. We mechanically mixed the powders to obtain a ratio of 70% olivine and 30% orthopyroxene by weight. The samples were dried at ~60 °C for several days before use in each experiment.

**Experimental procedure.** A schematic diagram of the sample assembly is shown in Supplementary Fig. 1. For each experiment, ~0.1 g of olivine-orthopyroxene powder, with 4 wt% deionized water added, was placed between two alumina forcing blocks (shear pistons) pre-cut at an angle of 45° with respect to the maximum compression direction. Grooves of ~150 μm depth and ~800 μm spacing were produced on the surfaces of the forcing blocks to ensure coupling at the piston–sample interface. The piston–sample assembly was placed inside a mechanically sealed inner Ag jacket (wall thickness of 0.1 mm) using a Pt disc at each end of the sample, and then slid into an outer Ni jacket (wall thickness of 0.2 mm) that slightly overlapped the end pistons (alumina). NaCl and talc were used as the confining medium inside and outside the graphite furnace, respectively. Temperature was measured using a Pt–Pt10%Rh thermocouple placed close to the centre of the Ni jacket.

All experiments were conducted at a temperature of 500 °C and a confining pressure of 1.0 GPa, using a Griggs-type solid-medium apparatus installed at Tohoku University, Japan. Pressure was initially raised to 700 MPa before pressure and temperature were simultaneously increased to the desired values over 1.5 h. The samples were then annealed at the desired pressure and temperature for 11–21 h, while the piston was advanced to come into contact with the sample. The samples were deformed at a constant axial displacement rate of $1.7 \times 10^{-4}$ to $3.3 \times 10^{-6}$ mm s$^{-1}$, corresponding to a shear strain rate of $2.2 \times 10^{-4}$ to $4.3 \times 10^{-6}$ s$^{-1}$. Shear strain was calculated from the axial displacement and the initial thickness of the sample (~0.6 mm). Two hydrostatic experiments of 20 and 168 h durations were performed for comparison. At the end of the experiments, the samples were quenched under load to 200 °C in ~5 min. Pressure and temperature were then lowered to ambient conditions over a period of 90 min.

Axial load and axial displacement were measured using an external load cell and a displacement transducer, respectively. Force–displacement data were corrected to account for changes in sample/shear piston overlap (assuming constant volume and homogeneous deformation), apparatus distortion, and friction due to advancement of the σ$_1$ piston. The baseline for friction correction was given by a linear extrapolation of the initial load increase prior to the hit point[40], which is the point at the intersection of the baseline and a linear fit to data on elastic loading of the sample (Supplementary Fig. 2). We also note that our experiments were conducted under undrained conditions and that pore fluid pressures within the Ag jacket were not constant throughout the experiment; instead, they decreased with increasing time and reaction progress[41].

**Analytical methods.** The recovered sample was cut into two sections (parallel to the shear direction and perpendicular to the shear plane) and its microstructure was investigated using SEM Major element analyses of minerals were performed using an electron probe microanalyzer (EPMA; JEOL JXA-8200 and JXA-8900 R) at Tohoku University and Shizuoka University, Japan, respectively. The analytical conditions were a 15–20 kV accelerating voltage, 12 nA beam current, and 1–5 μm beam size. Hydrous phyllosilicates were identified using a laser Raman spectrometer (HORIBA XploRA PLUS) at Tohoku University, Japan. The spectrum resolution was 1.0 cm$^{-1}$. A microscope with a ×100 objective lens was used to focus the incident laser beam (532 nm line of a green laser) into a 1 μm spot size and to collect the Raman signal from the sample. Spectra were acquired over 25–150 s of measurement time with the laser power of 10 mW. Representative Raman spectra in the range of 150–1150 cm$^{-1}$ are shown in Fig. 6.

**Data availability.** The data that support the findings of this study are available from the corresponding author on request.

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

## Acknowledgements

We acknowledge Masaoki Uno and Ryosuke Oyanagi for technical assistance in using the electron probe microanalyzer and laser Raman spectrometer, and Katsuyoshi Michibayashi for help in performing SEM analyses. We thank Shintaro Azuma, Keishi Okazaki and Takayuki Nakatani for helpful comments and discussions. This study was funded by a Grant-in-Aid for Young Scientists (B) (no. 25800279) and a Grant-in-Aid for Scientific Research on Innovative Areas (no. 26109005).

## Author contributions

K.-i.H. and J.M. designed the study, K.F., M.K. and J.M. conducted the deformation experiments, and K.F. and A.O. performed the chemical analyses. K.H. wrote the initial draft of the manuscript. All authors discussed the results and implications.

## Additional information

**Competing financial interests:** The authors declare no competing financial interests.

