## [Peer Review File · Nature Communications]

Reviewers' comments:

Reviewer #1 (Remarks to the Author):

This is an interesting paper which maybe of interest to a broader audience. Understanding weakening mechanism of rocks and minerals that can lead to reduction of strength required to initiate subduction is a fundamental and largely unsolved problem. This study contributes to that goal.

I have some comments that should be addressed.

1. The friction experiments start with peridotite gouge and that it is from this point which the new weakening processes are studied. However, the existence of peridotite gouge means that a localized fault zone has already developed tectonically and that this study really avoids the initial phase of localization and weakening.
2. Even when the gouge has been weakened by strain and reactions, the shear strength is still larger than the 30 MPa required to initiate subduction. Some more discussion on the range of strengths expected is needed and if these really how enough to initiate subduction.
3. The strain rates in these experiments are many orders of magnitude larger than natural settings. Do such unnatural strain rates limit the applicability to natural settings?
4. The conclusion, "Our findings suggest that infiltration of seawater into oceanic faults is an important controlling factor on the initiation of plate tectonics on terrestrial planets", is a common, generic conclusion which has been made repeatedly in the literature. Why does such a generic finding now merit publication in a high impact journal?

Reviewer #2 (Remarks to the Author):

This is an interesting and important paper demonstrating dramatic reduction of frictional strength of deforming mantle rocks due hydration reactions forming weak hydrous phyllosilicates (especially talc) in presence of water fluid. It is, therefore, suggested that deep infiltration of water into oceanic faults is an important controlling factor for initiation of subduction and plate tectonics. Whereas I generally agree with this conclusion it would be important to give more in-depth discussion on the processes of oceanic plates faulting and deep water infiltration. Figure 1 suggests that water infiltration should reach depths of >20-30 km into relatively cold plate interior where phyllosilicates should be thermodynamically stable. Such cold and thick plate conditions do not seem to correspond to shallow hydrothermal circulation of water and normal/detachment faulting (<10-12 km, e.g., Escartin et al., 2008) at hot mid-ocean ridges. One potential candidate process could be normal faulting related to oceanic plate bending that causes deep water suction at outer rise regions e.g. Ranero et al., 2003; Faccenda et al., 2009), but significance of similar

process for the oceanic plate interior needs to be demonstrated. In addition, according to the Anderson theory of faulting normal faults do not have optimal orientation (too steep) for their inversion into subduction thrusts, which may reduce efficiency of the strength lowering for subduction initiation. Possibly some additional rheological weakening mechanisms such as shear heating (e.g., Lu et al., 2015) and brittle/plastic strength lowering by high pore pressure of water (e.g., Dymkova and Gerya, 2013) released by reactivated faults dehydration may help but this needs to be evaluated.

Taras Gerya, Zurich 16.02.2016

Minor points

Fig.1. Meaning of d (grain size?, units?) needs to be explained.

References

Dymkova, D., Gerya, T. (2013) Porous fluid flow enables oceanic subduction initiation on Earth. *Geophysical Research Letters*, 40, 5671-5676.

Escartin, J., Smith, D.K., Cann, J., Schouten, H., Langmuir, C.H., Escrig, S., 2008. Central role of detachment faults in accretion of slow-spreading oceanic lithosphere. *Nature* 455, 790-795.

Faccenda, M., Gerya, T.V., Burlini, L. (2009) Deep slab hydration induced by bending related variations in tectonic pressure. *Nature Geoscience*, 2, 790-793.

Lu, G., Kaus, B.J.P., Zhao, L., Zheng, T. (2015) Self-consistent subduction initiation induced by mantle flow. *Terra Nova* 27 (2), 130-138

Ranero, C. R., Phipps Morgan, J., Reichert, C. (2003) Bending-related faulting and mantle serpentinization at the Middle America trench. *Nature* 425, 367-373.

Reviewer #3 (Remarks to the Author):

Review of the manuscript: Reaction-induced rheological weakening enables oceanic plate subduction.

This paper presents the results of a series of new experiments looking at the effect of hydration reactions on the rheological properties of ultramafic rocks, with relevance for the initiation of subduction. In brief, the experiments demonstrate that hydration of orthopyroxene-bearing ultramafic rocks will cause the pyroxene to react to form weak talc at 500°C. This reaction-induced weakening can explain the amount of weakening that apparently is required to initiate subduction, according to several studies cited at the beginning. The experiments are new and interesting, and generate talc at P-T-compositional conditions that, as far as I know, have not been tried previously. The paper is well written, and the English is very good. References cited appear to be appropriate and adequate. The results certainly are worth publishing, with some relatively minor changes/additions.

The authors describe one possible way to generate low-strength conditions on a fault that could make the fault a favorable site for the initiation of subduction. But there are other possibilities, such as fluid overpressures, which was suggested by Gerya et al. (2008), among others. There should be some

discussion of alternative weakening mechanisms.

There also is no mention of what the fluid pressure in the sample might be or how it might evolve over the length of the experiments. What might the effective confining pressure have been? Can one estimate the starting porosity to get some idea of at least the initial fluid pressure? The greater weakening at the slower strain rates (compared at a shear strain of 4) certainly is consistent with reaction weakening; nevertheless, the added water might have some effect on the measured shear stresses in addition to that caused by the growth of talc.

Table S2 lists an experiment using 100% orthopyroxene, run at the slowest strain rate. However, the final (slightly extrapolated) shear stress is more than a factor of 5 greater than that for the equivalent experiment using the 70:30 olivine-pyroxene mixture. What happened with that sample? Although the weakening reaction listed on lines 83-84 only involves the pyroxene as a reactant, the presence of olivine in the starting material would seem to be somehow required. Or was there something strange about that experiment?

The methods used to determine the proportion of talc in shears in the run products, plotted in Fig. S6, need to be described. Presumably, some image-analysis techniques were used, estimating the proportion of the darker-gray material in images such as Fig. 4E? How were the shear zone boundaries defined? Are there other minerals or materials with similar grayscale colors that could affect the results? Why wasn't the highest strain-rate experiment included in Fig. S6? The total strain for that experiment was a little lower than for the experiments that were plotted in Fig. S6, but the result would nevertheless be very interesting, e.g., for the discussion of the stress-strain data in Fig. 2. Also, the sentence on lines 98-99 in the main text, that mentions Fig. S6, does not make sense. I think that the authors wanted to say that all of the samples deformed at strain rates lower than 10^{-4} had relatively similar talc contents along the subsidiary shears. Is there an explanation for the amount of reaction correlating with strain rather than time? (Actually, there does seem to be at least a slight increase in % talc with decreasing strain rate -- increasing time -- in Fig. S6.) One more thing: the high-magnification SEM photos of the talc in Fig. 4 do not give any indication (to me, at least) that they were actually involved in shearing. Are there any places where one can see the talc aligned/smear out in the shearing direction?

Minor points (typos, etc.):

Line 43. Amphiboles are not phyllosilicate minerals. There are very few published strength data on amphiboles, but I would not expect them to be particularly weak.

Lines 89-90. The statement that: "...excess iron released during the formation of talc is accommodated by olivine via fluid migration" seems to be not quite right, if I interpret it correctly. I read it as pre-existing olivine taking up the excess Fe. However, if the stated reaction (lines 83-84) is correct, then the olivine is newly crystallized, preferentially as overgrowths on the preexisting olivine.

The Fig. 3 caption should include some mention of the added data at $\sim 1.8 \times 10^{-5}$ strain rate (the 5- and 6-sided polygons), given that they are not presented in Fig. 2.

Fig. 4 caption, last sentence. "iron-rich rims" (not iron-riched).

Response to reviewers

Reviewer #1:

This is an interesting paper which maybe of interest to a broader audience. Understanding weakening mechanism of rocks and minerals that can lead to reduction of strength required to initiate subduction is a fundamental and largely unsolved problem. This study contributes to that goal.

I have some comments that should be addressed.

1. The friction experiments start with peridotite gouge and that it is from this point which the new weakening processes are studied. However, the existence of peridotite gouge means that a localized fault zone has already developed tectonically and that this study really avoids the initial phase of localization and weakening.

Yes, we assumed that a localized fault zone has already developed along preexisting oceanic faults such as transform faults. This assumption has been utilized by numerous geodynamic models for subduction to be initiated (e.g., Hall et al., 2003; Gerya et al., 2008).

In the revised manuscript, we added the following text in the section of the Discussion (Lines 175 to 178), to explain how peridotite fault gouge develops due to aging of the lithosphere.

“Therefore, we suggest that as oceanic lithosphere ages, the depth extent of transform faults gradually increases via repeated brittle failures, resulting in the formation of phyllosilicate-rich peridotite fault gouge due to seawater infiltration and hydrothermal alteration.”

2. Even when the gouge has been weakened by strain and reactions, the shear strength is still larger than the 30 MPa required to initiate subduction. Some more discussion on the range of strengths expected is needed and if these really how enough to initiate subduction.

Our experimental results indicate that the highest strength at the brittle–ductile transition (BDT) decreases down to a few tens of megapascals. This means that the average strength of the entire fault must be lower than the strength at the BDT, as thermally activated creep results in a marked reduction in strength at depths below the BDT. We added the following text in the section of the Discussion (Lines 139 to 141).

“It is also important to note that thermally activated processes, such as dislocation and diffusion creep, become active at depths below the BDT (Fig. 1), thereby strongly reducing the plastic strength with

increasing temperature.”

3. The strain rates in these experiments are many orders of magnitude larger than natural settings. Do such unnatural strain rates limit the applicability to natural settings?

If we conduct our experiments at strain rates corresponding to natural conditions (e.g., 10^{-10} s^{-1}), the steady-state stress will be lower than that at our present strain rates, as talc has a positive dependence of μ_{ss} on sliding velocity (i.e., $a-b > 0$). In the revised manuscript, we added the following text in Lines 134 to 139.

“The minimum value of steady-state shear stress observed in our experiments (i.e., at $\dot{\gamma} = 4.3 \times 10^{-6} \text{ s}^{-1}$) was around 40 MPa (Fig. 2), which is slightly higher than the average shear stress required for subduction initiation on an incipient thrust fault^{7,8}. However, friction experiments have shown that the steady-state friction coefficient (μ_{ss}) of talc has a positive dependence on sliding velocity V ($\Delta\mu_{\text{ss}}/\Delta\ln V = 0.001\text{--}0.008$ at 400°C)¹⁶, suggesting a further decrease in stress at natural strain rates.”

4. The conclusion, "Our findings suggest that infiltration of seawater into oceanic faults is an important controlling factor on the initiation of plate tectonics on terrestrial planets", is a common, generic conclusion which has been made repeatedly in the literature. Why does such a generic finding now merit publication in a high impact journal?

In the revised manuscript, we substantially modified the conclusion to give more in-depth discussion on oceanic subduction initiation induced by seawater infiltration into the upper mantle (see Lines 166 to 178).

Reviewer #2:

This is an interesting and important paper demonstrating dramatic reduction of frictional strength of deforming mantle rocks due hydration reactions forming weak hydrous phyllosilicates (especially talc) in presence of water fluid. It is, therefore, suggested that deep infiltration of water into oceanic faults is an important controlling factor for initiation of subduction and plate tectonics.

Whereas I generally agree with this conclusion it would be important to give more in-depth discussion on the processes of oceanic plates faulting and deep water infiltration. Figure 1 suggests that water infiltration should reach depths of >20-30 km into relatively cold plate interior where phyllosilicates should be thermodynamically stable. Such cold and thick plate conditions do not seem to correspond to shallow hydrothermal circulation of water and normal/detachment faulting (<10-12 km, e.g., Escartin et al., 2008) at hot mid-ocean ridges. One potential candidate process could be normal faulting related to oceanic plate bending that causes deep water suction at outer rise regions e.g. Ranero et al., 2003; Faccenda et al., 2009), but significance of similar process for the oceanic plate interior needs to be demonstrated. In addition, according to the Anderson theory of faulting normal faults do not have optimal orientation (too

steep) for their inversion into subduction thrusts, which may reduce efficiency of the strength lowering for subduction initiation.

We agree that the extent of oceanic transform faults is limited to several kilometers depth at most. However, several seismic studies have shown that the long, slow-slipping transform faults (e.g., Romanche) on slow-spreading mid-ocean ridges can extend deeper into the mantle (Abercrombie and Ekström, 2001; Roland et al., 2010), because lower thermal gradients allow brittle failure even at mantle depths. Therefore, we believe that water–rock interaction can lead to the development of weak, talc-bearing peridotite gouge, which provide a favorable condition for oceanic plate subduction.

In the revised manuscript, we added the following text in the section of the Discussion (Lines 166 to 175).

“The mechanism that introduces water into the upper mantle within oceanic lithosphere remains unclear, because preexisting weak zones (such as transform faults and fracture zones) are commonly limited to several kilometers beneath the seafloor^{2,33,34}. However, this is not always the case: for example, in the present day, the long (900 km), low-slip-rate (2 cm yr^{-1}) Romanche Fault, a transform fault on the Mid-Atlantic Ridge, is characterized by seismicity that extends to depths of 20 km, indicating that the fault zone penetrates into the upper mantle³⁵. For a half-space cooling model, the temperature at the center of the fault at 20 km depth is $\sim 500^\circ\text{C}$ ³⁵, which lies within the stability field of talc ($<700^\circ\text{C}$ ³⁶). Geodynamic models have also predicted that the depths of brittle deformation and fluid circulation increase with decreasing geothermal gradient³⁷.”

Although bending-related deep faulting and mantle hydration will occur near the trenches (Ranero et al., 2003, *Nature*), such processes can happen only after subduction of the oceanic plate starts. This is the chicken-and-egg problem, as suggested by Korenaga (2013, *Annu. Rev. Earth Planet. Sci.*). Therefore, we do not consider the possibility that deep outer rise faults act as the preexisting zone for subduction initiation.

Possibly some additional rheological weakening mechanisms such as shear heating (e.g., Lu et al., 2015) and brittle/plastic strength lowering by high pore pressure of water (e.g., Dymkova and Gerya, 2013) released by reactivated faults dehydration may help but this needs to be evaluated.

In the revised manuscript, we cited Lu et al. (2015) and Dymkova and Gerya (2013) in Lines 159 to 161, where we state there are alternative weakening mechanisms such as shear heating and high pore fluid pressures, in addition to our model. We also mentioned that our experimental results support the maintenance of high pore fluid pressures in deep transform faults, as the development of phyllosilicate (i.e., talc) would lead to a marked decrease in permeability (see Lines 161 to 165).

Taras Gerya, Zurich 16.02.2016

Minor points

Fig.1. Meaning of d (grain size?, units?) needs to be explained.

We explained the meaning of d (grain size) in the legend of Figure 1 (Lines 345 to 346).

References

- Dymkova, D., Gerya, T. (2013) Porous fluid flow enables oceanic subduction initiation on Earth. Geophysical Research Letters, 40, 5671-5676.
- Escartin, J., Smith, D.K., Cann, J., Schouten, H., Langmuir, C.H., Escrig, S., 2008. Central role of detachment faults in accretion of slow-spreading oceanic lithosphere. Nature 455, 790-795.
- Faccenda, M., Gerya, T.V., Burlini, L. (2009) Deep slab hydration induced by bending related variations in tectonic pressure. Nature Geoscience, 2, 790-793.
- Lu, G., Kaus, B.J.P., Zhao, L., Zheng, T. (2015) Self-consistent subduction initiation induced by mantle flow. Terra Nova 27 (2), 130-138
- Ranero, C. R., Phipps Morgan, J., Reichert, C. (2003) Bending-related faulting and mantle serpentinization at the Middle America trench. Nature 425, 367-373.

Reviewer #3:

Review of the manuscript: Reaction-induced rheological weakening enables oceanic plate subduction.

This paper presents the results of a series of new experiments looking at the effect of hydration reactions on the rheological properties of ultramafic rocks, with relevance for the initiation of subduction. In brief, the experiments demonstrate that hydration of orthopyroxene-bearing ultramafic rocks will cause the pyroxene to react to form weak talc at 500{degree sign}C. This reaction-induced weakening can explain the amount of weakening that apparently is required to initiate subduction, according to several studies cited at the beginning. The experiments are new and interesting, and generate talc at P-T-compositional conditions that, as far as I know, have not been tried previously. The paper is well written, and the English is very good. References cited appear to be appropriate and adequate. The results certainly are worth publishing, with some relatively minor changes/additions.

The authors describe one possible way to generate low-strength conditions on a fault that could make the fault a favorable site for the initiation of subduction. But there are other possibilities, such as fluid overpressures, which was suggested by Gerya et al. (2008), among others. There should be some discussion of alternative weakening mechanisms.

We added the following text in the section of the Discussion (Lines 159 to 165), to discuss alternative mechanisms to truncate the brittle–plastic strength envelope.

“On the other hand, high pore fluid pressures^{5,30} and shear heating³¹ have been suggested as possible mechanisms that truncate the brittle–plastic strength envelope. In particular, an increase in pore fluid pressure (P_f) within a fault zone acts to reduce the effective normal stress ($\sigma_n^{\text{eff}} = \sigma_n - P_f$) and thereby lower the shear stress. Our observations may imply that the development of interconnected networks of phyllosilicates (Fig. 5b,c), which have low permeabilities³², allows high P_f values to persist in the deep

portions of oceanic faults (if the input of fluids is continuous).”

There also is no mention of what the fluid pressure in the sample might be or how it might evolve over the length of the experiments. What might the effective confining pressure have been? Can one estimate the starting porosity to get some idea of at least the initial fluid pressure? The greater weakening at the slower strain rates (compared at a shear strain of 4) certainly is consistent with reaction weakening; nevertheless, the added water might have some effect on the measured shear stresses in addition to that caused by the growth of talc.

Unfortunately, we do not have the way to estimate the effective confining pressure (P_c^{eff}) in the sample through the experiments, because we cannot accurately measure it after the application of confining pressure because the sample is enclosed in the metal jacket. Indeed, this is the fundamental problem in conducting hydration or dehydration experiments using solid-medium deformation apparatuses (e.g., Chernak et al., 2009, *JGR*; Getsinger and Hirth, 2014, *Geology*; Proctor and Hirth, 2015, *EPSL*; Okazaki and Hirth, 2016, *Nature*). We also agree that the values of P_c^{eff} within the sample evolve with increasing time because the fluid escapes from the Ag capsule and the formation of talc is OH-consuming reaction. However, we infer that the P_c^{eff} by added water might only affect the initial strength before the reaction starts and it gradually decreases due to the talc formation. In the revised manuscript, we added the following text in the section of the Methods (Lines 219 to 222).

“We also note that our experiments were conducted under undrained conditions and that pore fluid pressures within the Ag jacket were not constant throughout the experiment; instead, they decreased with increasing time and reaction progress³⁹.”

Table S2 lists an experiment using 100% orthopyroxene, run at the slowest strain rate. However, the final (slightly extrapolated) shear stress is more than a factor of 5 greater than that for the equivalent experiment using the 70:30 olivine-pyroxene mixture. What happened with that sample? Although the weakening reaction listed on lines 83-84 only involves the pyroxene as a reactant, the presence of olivine in the starting material would seem to be somehow required. Or was there something strange about that experiment?

In the revised manuscript, we added the stress–strain data for the pure orthopyroxene (opx) gouge in Supplementary Figure 3. The figure shows that in case of 100% opx, a steady-state shear stress was not attained even at $\gamma \sim 4$, possibly implying that the presence of olivine (ol) in the starting material seems to be somehow required, as pointed out by Reviewer #2. However, the present experiments using the 70:30 ol-opx mixture models the Earth’s upper mantle. To test the hypothesis on the role of olivine for the reaction, new experiments using the ol/opx samples with different opx contents need to be conducted. However, we think of that is beyond the focus of this study.

The methods used to determine the proportion of talc in shears in the run products, plotted in Fig. S6, need

to be described. Presumably, some image-analysis techniques were used, estimating the proportion of the darker-gray material in images such as Fig. 4E? How were the shear zone boundaries defined? Are there other minerals or materials with similar grayscale colors that could affect the results?

Yes, we conducted image analysis on polished sections, using the back-scattered electron (BSE) images. We also confirmed that there are no other minerals with similar grayscale colors. We added the following text in the legend of Figure 7 (Lines 379 to 382).

“The area fraction of talc within $\sim 50 \mu\text{m}$ of the localized shears was deduced from image analysis, using back-scattered electron (BSE) imaging. From each BSE image, we obtained the area fractions of pores (black), olivine (right grey), and talc (dark grey).”

Why wasn't the highest strain-rate experiment included in Fig. S6? The total strain for that experiment was a little lower than for the experiments that were plotted in Fig. S6, but the result would nevertheless be very interesting, e.g., for the discussion of the stress-strain data in Fig. 2.

In the revised manuscript, we newly added the data of talc proportion for the highest strain rate experiment in Figure 7. We thought that rate-insensitive deformation was predominant at the highest strain rate experiment (Lines 116 to 121).

Also, the sentence on lines 98-99 in the main text, that mentions Fig. S6, does not make sense. I think that the authors wanted to say that all of the samples deformed at strain rates lower than 10^{-4} had relatively similar talc contents along the subsidiary shears. Is there an explanation for the amount of reaction correlating with strain rather than time? (Actually, there does seem to be at least a slight increase in % talc with decreasing strain rate – increasing time – in Fig. S6.)

We agree that the trend of a slight increase in talc proportion can be recognized. We added the following text in Lines 110 to 111.

“At $\dot{\gamma} < 10^{-4} \text{ s}^{-1}$, the average proportion of talc formed along the localized shears increases only slightly (from 25% to 30%; Fig. 7) with decreasing shear strain rate,”

However, although Runs KF-08 and KF-09 are done by the similar total times (sum of heating and deformation times) of 24 and 23 hours (Table 2), respectively, there is a marked contrast in talc proportion (see Figure 5a and 5b), which depends on total shear strains. We therefore suggest that the amount of strain is the primary control on reaction progress.

In the revised manuscript, we added the following text in the section of the Discussion (Lines 112 to 116).

“We also find that larger shear strains lead to a greater amount of reaction products, despite a total heating time that is almost constant (~ 24 h; compare Fig. 5a with Fig. 5b). These results indicate that the reaction rate of the fault zone material depends on the amount of strain that accumulates during cataclastic

deformation, rather than time.”

One more thing: the high-magnification SEM photos of the talc in Fig. 4 do not give any indication (to me, at least) that they were actually involved in shearing. Are there any places where one can see the talc aligned/smeared out in the shearing direction?

Yes, we have found the sites where talc aligns along the localized shear zones (e.g., B or R₁ shears). In the revised manuscript, we added a representative high-magnification SEM image showing aligned talc grains along the shear plane in Figure 5e. Instead, Figure 5e and 5f in the previous manuscript are moved to Figure 5f and Supplementary Figure 4, respectively, in the revised manuscript. We also added the following text in the section of the Results (Lines 86 to 87).

“The basal plane of the talc grains is strongly aligned with the localized shear plane (Fig. 5e).”

Minor points (typos, etc.):

Line 43. Amphiboles are not phyllosilicate minerals. There are very few published strength data on amphiboles, but I would not expect them to be particularly weak.

We replaced ‘hydrous phyllosilicates’ with ‘hydrous minerals’ (Lines 39 to 40). We also added the following text in Lines 40 to 41.

“In particular, serpentine and talc as phyllosilicates have much lower friction coefficients than anhydrous mantle minerals^{15,16}.”

Lines 89-90. The statement that: "...excess iron released during the formation of talc is accommodated by olivine via fluid migration" seems to be not quite right, if I interpret it correctly. I read it as pre-existing olivine taking up the excess Fe. However, if the stated reaction (lines 83-84) is correct, then the olivine is newly crystallized, preferentially as overgrowths on the preexisting olivine.

We modified the statement of the reaction involving excess Fe to state that Fe-rich olivine was newly crystallized on the preexisting olivine (see Lines 100 to 102).

The Fig. 3 caption should include some mention of the added data at $\sim 1.8 \times 10^{-5}$ strain rate (the 5- and 6-sided polygons), given that they are not presented in Fig. 2.

We added the following text in the legend of Figure 4 in the revised manuscript (Lines 358 to 360).

“Pentagonal and hexagonal symbols indicate experiments in which the shear deformation was stopped at shear strains (γ) of 1.1 and 2.7, respectively (not shown in Figure 2).”

Fig. 4 caption, last sentence. "iron-rich rims" (not iron-riched).

We replaced 'iron-riched' with 'iron-rich' (Line 401).

Reviewers' comments:

Reviewer #2 (Remarks to the Author):

The paper has been considerably improved by revisions. One new point, however, needs a discussion. The authors suggest that the investigated weakening mechanism explain plate tectonics initiation along transform faults.

"Our findings suggest that infiltration of seawater into transform faults with long lengths and low slip rates is an important controlling factor on the initiation of plate tectonics on terrestrial planets,"

The critical question is, however, how transform faults can develop before plate tectonics? At present day conditions transform faults are consequences of the ongoing plate tectonics. On the other hand, Harris and Bedard (2015) discussed formation of strike-slip zones (as well as rifts and indentation zones) on Venus by mantle-flow-induced movements of cratonic-like domains in the absence of global plate tectonics. In addition, Bercovici and Ricard (2014) proposed formation of transform plate boundaries by restraining of other boundary types (subduction zones, ridges) due to their long-term weakness induced by damage. I understand that the authors are at the limit with the number of references but I still think that the origin of transform faults before plate tectonics is an important question directly related to one of their main conclusions.

Taras Gerya, Zurich 18.04.2016

References

Bercovici, D. & Ricard, Y., 2014. Plate tectonics, damage and inheritance. *Nature* 508, 513-516.

Harris, L. B., Bédard, J. H., 2015. Interactions between continent-like 'drift', rifting and mantle flow on Venus: gravity interpretations and Earth analogues. *Geological Society, London, Special Publications* 401 (1), 327-356.

Reviewer #3 (Remarks to the Author):

Comments on the revised manuscript: Reaction-induced rheological weakening enables oceanic plate subduction".

I have read over all of the reviewers' comments and the point-by-point responses made by the authors. I also reread the full manuscript.

I feel that the authors did a good job of addressing all of the reviewers' comments, and the manuscript should be accepted. The only suggestion I would make is to remove the 2-line section of chemical data

(lines 53-54). The starting material (olivine and orthopyroxene) compositions can be mentioned in the sentence on lines 56-59, for example: "...on hot-pressed aggregates of olivine (70%) and orthopyroxene (30%) of initial compositions Fo90 and En90, respectively, at a temperature..." The olivine rim compositions are already noted elsewhere.

Response to reviewers

Reviewer #2:

The paper has been considerably improved by revisions. One new point, however, needs a discussion. The authors suggest that the investigated weakening mechanism explain plate tectonics initiation along transform faults.

"Our findings suggest that infiltration of seawater into transform faults with long lengths and low slip rates is an important controlling factor on the initiation of plate tectonics on terrestrial planets,"

The critical question is, however, how transform faults can develop before plate tectonics? At present day conditions transform faults are consequences of the ongoing plate tectonics. On the other hand, Harris and Bedard (2015) discussed formation of strike-slip zones (as well as rifts and indentation zones) on Venus by mantle-flow-induced movements of cratonic-like domains in the absence of global plate tectonics. In addition, Bercovici and Ricard (2014) proposed formation of transform plate boundaries by restraining of other boundary types (subduction zones, ridges) due to their long-term weakness induced by damage. I understand that the authors are at the limit with the number of references but I still think that the origin of transform faults before plate tectonics is an important question directly related to one of their main conclusions.

In the revised manuscript, we discussed how transform faults develop prior to the operation of plate tectonics (see Lines 189 to 203), as suggested. We mentioned that strike-slip faults have been found in stagnant-lid tectonics (i.e., Venus), by citing Harris and Bédard (2014). We also cited Bercovici and Ricard (2014) to suggest that damage in the presence of seawater lead to the development of talc-bearing weak zones in the lithosphere, as seen in our experiments.

Taras Gerya, Zurich 18.04.2016

References

Bercovici, D. & Ricard, Y., 2014. Plate tectonics, damage and inheritance. Nature 508, 513-516.

Harris, L. B., Bédard, J. H., 2015. Interactions between continent-like 'drift', rifting and mantle flow on Venus: gravity interpretations and Earth analogues. Geological Society, London, Special Publications 401 (1), 327-356.

Reviewer #3 (Remarks to the Author):

Comments on the revised manuscript: Reaction-induced rheological weakening enables oceanic plate subduction".

I have read over all of the reviewers' comments and the point-by-point responses made by the authors. I also reread the full manuscript.

I feel that the authors did a good job of addressing all of the reviewers' comments, and the manuscript should be accepted. The only suggestion I would make is to remove the 2-line section of chemical data (lines 53-54). The starting material (olivine and orthopyroxene) compositions can be mentioned in the sentence on lines 56-59, for example: "...on hot-pressed aggregates of olivine (70%) and orthopyroxene (30%) of initial compositions Fo_{90} and En_{90} , respectively, at a temperature..." The olivine rim compositions are already noted elsewhere.

We deleted the section of chemical data as suggested. We also added the following text in the section of mechanical data (Lines 66 to 68).

"Simple-shear deformation experiments were conducted on hot-pressed aggregates of olivine (70%) and orthopyroxene (30%) of initial compositions Fo_{91} and En_{90} , respectively (Table 1),"